# On the (Apparently) Paradoxical Role of Noise in the Recognition of Signal Character of Minor Principal Components

Alessandro Giuliani [1,*] and Alessandro Vici [2]

[1] Environment and Health Department, Istituto Superiore di Sanità, Viale Regina Elena 299, 00161 Rome, Italy

[2] Department of Oncology and Molecular Medicine, Istituto Superiore di Sanità, Viale Regina Elena 299, 00161 Rome, Italy; vici.1769239@studenti.uniroma1.it

* Correspondence: alessandro.giuliani@iss.it

**Abstract:** The usual method of separating signal and noise principal components on the sole basis of their eigenvalues has evident drawbacks when semantically relevant information 'hides' in minor components, explaining a very small part of the total variance. This situation is common in biomedical experimentation when PCA is used for hypothesis generation: the multi-scale character of biological regulation typically generates a main mode explaining the major part of variance (size component), squashing potentially interesting (shape) components into the noise floor. These minor components should be erroneously discarded as noisy by the usual selection methods. Here, we propose a computational method, tailored for the chemical concept of 'titration', allowing for the unsupervised recognition of the potential signal character of minor components by the analysis of the presence of a negative linear relation between added noise and component invariance.

**Keywords:** principal component analysis; semantic information; noise; bioinformatics; hypothesis generation; unsupervised learning

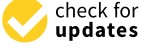



## 1. Introduction

### 1.1. The Peculiar Role of Principal Component Analysis in Life Sciences

Principal component analysis (PCA) is the most-widespread statistical method to deal with multidimensional data sets coming from biomedical research. The huge success of PCA is somewhat related to its location in a twilight zone between general-purpose statistical techniques and quantitative models developed for specific applications [1,2]. Even if any statistical method (e.g., Analysis of Variance) implies some general hypothesis on the analyzed phenomenon (e.g., linearity), these hypothesis pertain to a meta-level largely decoupled from the nature of the analyzed system. On the contrary, other approaches (e.g., the Renormalization Group, a formal apparatus that allows for systematic investigation of the changes of a physical system as viewed at different scales [3]) are linked to a specific field of investigation and allow only for a 'metaphorical' extension to other science areas [4]. PCA blurs the distinction between 'content-agnostic' and 'content-derived' methods: while being a general-purpose statistical technique, PCA allows generating a statistical mechanics theoretical frame for biological systems without the need of strong a priori theoretical assumptions [2]. This is made possible by the analogy of PCA to the two main pillars of systems approaches in biomedicine: complex networks and order/organization quantification. In the first case, the analogy stems from the equivalence between a network adjacency matrix (with nodes corresponding to variables and edges to their pairwise correlations) and a correlation (or covariance) matrix [5]. In the second case, the analogy builds upon the immediate translation of the eigenvalues' distribution in terms of the order and the organization of the system at hand [6–9]. In both cases, PCA acts as a hypothesis-generation tool; thus, it is of utmost importance to assign a biologically relevant meaning to the extracted components. This feature contrasts with hypothesis testing approaches (e.g., Analysis of Variance, Student's *t*-test), where the biological meaning is set before the statistical analysis.

In the following, we will concentrate on a proposal for recognizing biologically relevant signals 'hidden' in very minor (in terms of the amount of explained variance) noise, a very common situation encountered in biological experimentation [10].

It is important to stress that, not only in biology, the interpretation of component meaning is of crucial importance: Vilenchik and colleagues in their paper [11] put the problem in a very straightforward way, 'To interpret or not to interpret PCA? This is our question'. The paper [11] dealt with the features of social networks coming from different platforms (e.g., Linkedin). The authors collected between 11 and 15 features per network that describe the user's activity in the network and the feedback that a user receives from other users. They constructed a 'interpretability score' based on the intuitive idea that the component most easily interpretable is the one in which only one feature (original variable) has a relevant (non-zero in the paper, which sets a threshold for loading relevance) loading. On the contrary, the presence of non-zero loadings for all the variables points to a very low interpretability score. This approach resembles the classical Shannon definition of information entropy, low interpretability scores being strictly related to the high values of the eigenvalues' distribution entropy, which in turn was demonstrated [9] to be the one of the most-efficient complexity estimators.

### 1.2. Why in Biomedical Sciences the Relevant Information Often Hides in Minor Components

When applying PCA in a content-agnostic mode (i.e., without the need for interpreting the meaning of extracted components) as in de-noising or dimensionality reduction for feature extraction applications [12,13], the choice of $p < n$ (with $n$ = the number of original variables), the number of components to be retained for further analysis, relies on their relative proportion of explained variance. When the focus is on the biological interpretation of extracted components, relying exclusively on the relative amount of variance explained as the selection criterion can be highly misleading. The order of selection by the eigenvalue deals with the concept of the 'noise floor', i.e., the set of minor components corresponding to extremely low and almost identical proportions of explained variance [14] that are dominated by stochasticity and, consequently, do not carry relevant information about the system at hand [5,14]. While the above statement is surely correct, the peculiar character of biological systems often generates a different situation in which the most-important thing is to investigate the information carried by specific minor components making part of the noise floor. One of the most-common situations in the life sciences is the presence of a dominant first component carrying semantically irrelevant information together with a set of minor components, where noise and weak (but semantically relevant) signals are mixed. An exemplary case is gene expression data sets coming from the microarray methodology [7,10]. These are extremely high-dimensionality data sets made of thousands of different gene expression values, in which the first component is related to the tissue of origin [7]. This is a natural consequence of the need for each specific tissue to maintain a peculiar balance among different gene expressions to carry out its physiological role, in other words, the dominant component neatly emerging from the noise floor, in terms of the explained variance, conveying semantically irrelevant information, corresponding to the kind of analyzed biological material that the scientist knows in advance.

In general, the first component is the component ecologists and morphologists traditionally define as 'size' as opposed to 'shape' components [15]. A 'size' component shows loadings of the same sign for all the variables. This is because of the presence of a major order parameter (typically the size of an organism), influencing almost equally all the studied variables. In this case, the relevant information resides in the minor components, in which the presence of both negative and positive loadings corresponds to changes in the 'shape' (in the wide sense of the individual profile across the different descriptors) of the statistical units. The most-direct way to discriminate between a weak signal (a semantically informative shape component) and a 'pure noise' component is through a supervised approach: a weak signal component, at odds with a pure noise one, has a significant correlation with an external variable of interest (Roden).

In the presence of a relevant correlation between a minor component and an external variable of interest, it is straightforward to sketch a mechanistic hypothesis by the analysis of both component loadings and scores [10]. What can we do if we have no external variable acting as the probe? This is not pure imagination: in many situations (e.g., neuroimages [16]), we do not have any external reference to rely upon, but, despite that, it is of crucial importance to determine the signal character of a comparatively minor detail.

The proposed method faces the problem of determining the signal character of minor components when in the absence of external covariates, relying upon the different behavior of the signal and noise when the original data matrix is corrupted by increasing amounts of added noise.

## 2. Methods

*Noise–Signal Discrimination by Noise Titration*

In [17], the authors described a toy model of a situation characterized by a semantically irrelevant major component, two minor (weak, but informative) components, and two pure noisy (measurement error) components. This configuration stems from the aforethought choice of a poor measurement frame: 33 Europeans cities were described by their distances (manually estimated on a geographic map at a $1 : 3 \times 10^6$ scale by a ruler, with a 1 mm precision corresponding to 3 km) from the five main towns of Latium (the central Italian region around Rome). The aim of the study was to reconstruct the mutual positions of the European towns. The fallacy of the measurement frame came from the very low size of Latium with respect to Europe (the ratio between Latium and the European areas is equal to $1.65 \times 10^{-4}$). While the choice of reference points covering a space of the same-order-of-magnitude data set range of variation could guarantee a precise reconstruction of the statistical units' configuration [18], the reference frame of [17] generated a strongly biased configuration. PCA was applied to the data set spanned by the distances from the five Latium towns, generating a five-component solution with the following distribution of explained variance proportion (Table 1).

**Table 1.** The table reports the explained variance proportion of the components extracted in [17]. PC1 accounts for the, by far, major part of the variance, but does not convey any semantically relevant information. PC2 and PC3 are the minor, but semantically relevant components, able to reconstruct the European cities' configuration. PC4 and PC5 are pure noise components corresponding to measurement error.

| PC1 | PC2 | PC3 | PC4 | PC5 |
|---|---|---|---|---|
| 0.996 | 0.0029 | 0.00006 | 0.00004 | 0.000005 |

PC1 was a size component with all-positive and near-unity loading, corresponding to the distances of the 33 European towns from the geographic center of Latium (approximately coincident with Rome). PC1, even if almost entirely explaining the data set's variance, carries useless information for reconstructing the data set's configuration: the circular symmetry of the distance operator makes cities like Barcelona and Athens have very similar PC1 scores even if they are very far apart (Barcelona and Athens being at the opposite west and east sides with respect to Latium). Both PC2 and PC3 are shape components with both (near-zero) positive and negative loadings with the five original variables. Adopting a supervised approach, the authors demonstrated that these two components, despite their very low eigenvalues, allow reconstructing the Europe cities' spatial configuration: the bearing between European cities and Latium was predicted by a linear combination of the PC2 and PC3 scores with a Pearson $r = 0.97$ ($p < 0.0001$) [17].

On the contrary, PC4 and PC5 were pure noise components, derived from the measurement error, with no relation to any geographical information.

The eigenvalue distribution of this geographical data set was coincident with many biological data sets in which a dominant (but semantically irrelevant) signal 'squashes' on the noise floor the semantically relevant, but weak signals.

In the above-described paper, the authors, having demonstrated the informative character of PC2 and PC3 by a supervised approach (correlation with an external covariate), gave a proof of concept of the suitability for the identification of very weak signals, using an unsupervised approach based on a 'noise titration' procedure [19]. This implies the generation of $n$ 'dirty' copies of the original data matrix, adding to each variable a zero-mean Gaussian noise with different standard deviation values. As noise levels increase (the signal-to-noise ratio (SNR) decreases), the weak signal components, extracted from the noise-corrupted data sets, showed a linear negative relation between the amount of added noise and their Pearson correlation with the corresponding components of the original data set. On the contrary, the pure noise (PC4, PC5) components did not show any significant relation between the amount of added noise and their Pearson correlation with the corresponding components of the original data set.

Poon and Barahona [19] explicitly set the metaphor of 'noise titration' in the framework of deterministic chaos/stochasticity discrimination. The authors equated time series data to acid–base solutions with deterministic chaos being most 'acidic' and white noise most 'alkaline'. This metaphor implies the consideration of deterministic chaos as a 'buffer', i.e., a 'weak acid' able to neutralize the effect of the added alkaline substance (noise) with no $pH$ change of the solution until a given volume (called the volume of titration) is reached. At this point, the pH of the solution initiates a dramatic rise.

An essential component in chemical titration is a sensitive indicator that specifically reveals the changes in pH around the equivalence point of acid–base neutralization. The indicator for the numerical titration is not specified by the authors, who stated, '...could in theory be any noise-tolerant technique that can reliably detect nonlinear dynamics in short noisy series' [19].

Our proposal, even if tailored for the Poon and Barahona metaphor, is more general. We adopted a more-fundamental (not necessarily linked to buffering) definition of titration as a method to determine the concentration of any substance (analyte) in a solution. A reagent (a substance known to react with the analyte) is added at different (known) volumes to the solution containing the analyte. The volume of added reagent corresponding to reaching a plateau of the chemical reaction is termed the 'titration volume' and corresponds to the volume of the analyte present in the solution (and consequently, to its concentration). We did not focus on the reaching of a 'neutralization point' nor on a quantification of the relative amount of 'determinism' and 'stochasticity', but limited ourselves to the recognition of the most-promising 'signal candidates' among the noise floor components. We hypothesized that a weak signal component should show a strong negative relation between added noise and the Pearson correlation between its original and noisy versions. This relation is expected to be less evident in 'pure noise' components. This mirrors the results reported in [17], which, in a titration framework, correspond to the progressive approaching of the end of the 'reaction' between the signal and added noise when the analyte (signal) is exhausted by the reaction with the increasing amount of the reactant (added noise). Given that noise cannot react with itself, the 'reaction' ends at an amount of added noise corresponding to the erasure of any residual correlation structure, and the relation between the amount of added noise and a suitable 'signal indicator' (i.e., the correlation between the initial signal and its noise-corrupted counterpart) vanishes. The main difficulty for an immediate generalization of the case described in [17] is that PC4 and PC5 are pure material noise deriving from actual measurement and are neatly separated by weak signal components in terms of the proportion of explained variance (see Table 1), thus greatly enhancing the discrimination power of the procedure. Moreover, the clear-cut separation in terms of explained variance between the weak signal and pure noise components keeps the ordering of the components across noise-corrupted replicas

invariant, so allowing for an immediate recognition of the corresponding components among the different data sets.

Thus, to develop a selection procedure able to identify putative weak signals endowed with a wide application spectrum, we designed a simulation framework making this discrimination as difficult as possible. In the following, we will describe the subsequent steps of the suggested procedure, explaining the motivations for each step:

1.  Using a relatively short series:
    The first step of our simulation is the generation of a time series ($S0$) by a Gaussian distribution at zero mean and unit standard deviation $N(0, 1)$. This series, by the action of a 15-dimensional embedding procedure at $lag = 1$ [14], gives rise to a multivariate matrix ($A0$) with 15 variables and 86 statistical units. The relatively low number of statistical units is consistent with the numerosity of a great part of biological experimentation.

2.  Presence of correlation structures in the extracted components:
    The character of PCA as a filter for correlated information makes the component scores relative to the $A0$ matrix show a certain amount of internal correlation, so we do not have any 'pure noise' component. In the second step of the procedure, we added to the original series an extremely weak signal (corresponding to a 'saw-wave'), composed of alternating 0.1 and $-0.1$ values. The extremely low power of the added signal makes the resulting series ($S1$) practically identical to $S0$ (Pearson $r = 0.994$; see Section 3) and have a largely superimposable eigenvalue distribution of the original ($A0$) and signal-added ($A1$) embedding matrices (see Section 3). Looking at component loading matrix, the 14th component, well inside the noise-floor, keeps trace of the square-wave signal. This component is the 'analyte' we expect to 'react' with the added noise.

3.  Adding noise:
    Having checked the superposition between the $S0$ and $S1$ series and the consequent coincidence between the eigenvalue distribution of the relative $A0$ and $A1$ 15-dimensional embedding matrices, as the third step, we generated 20 noise-corrupted copies of $S1$. These contaminated series are named $C1$–$C20$ (according to the increasing amount of added zero-mean Gaussian noise), from a minimal $\sigma$ of 0.05 ($C1$) to a maximum of 1 ($C20$), with each copy differing 0.05 $\sigma$ units from the previous one.

4.  Titration:
    As the fourth step, the 15-dimensional embedding matrices relative to the $C1$–$C20$ series are analyzed by PCA and the Pearson correlation between the PC6 and PC15 (noise floor) component scores relative to the $A1$ matrix and each corresponding component relative to each of the 20 embedding matrices' noise-added series are computed. It is worth noting that, due to their almost identical eigenvalues, the PC6–PC15 ordering varies both across noise-corrupted data sets and with respect to the original $A1$ matrix. Thus, for any value of added noise, we picked up the component having the higher correlation with the original one as the 'corresponding component', independent of its relative order of explained variance.

5.  Recognition of the weak signal:
    The fifth and last step of the procedure is to check if the 'weak signal' component (PC14 in the original $A1$ matrix) shows a significantly higher $R^2$ between the original corrupted versions of the PC14 (weak signal) correlation and the amount of added noise with respect to the other minor components. It is worth noting that the recognition of the weak signal relies on the expected effect of added noise in decreasing the correlation between the original and noise-contaminated versions of the component (the Pearson $r$ between added noise and original noise-added components are all negative).
    All the analyses were performed in R using the 'rnorm' function for randomly generating data from a normal distribution and the 'princomp' for PCA. Both functions belong to the 'stats' package.

## 3. Results and Discussion

The component loadings and eigenvalue distribution of the *A*1 (15-dimensional embedding of the *S*1 series) matrix is reported in Table 2.

**Table 2.** Explained variance for normal data distribution + low-amplitude signal. As expected in the case of a Gaussian distribution, the amount of explained variance gently decays along components with no clear-cut separation between the signal and noise floor. PC14's loading pattern partially mirrors the added low-amplitude saw-signal.

| Variables | Components | | | | | | | |
|---|---|---|---|---|---|---|---|---|
| | PC1 | PC2 | PC3 | PC4 | PC5 | PC6 | PC7 | PC8 |
| T0 | 0.293 | 0.000 | 0.443 | 0.000 | 0.191 | 0.000 | 0.259 | 0.000 |
| T1 | 0.307 | 0.000 | 0.000 | 0.447 | 0.141 | −0.272 | 0.175 | −0.130 |
| T2 | 0.180 | −0.167 | −0.168 | −0.105 | 0.473 | −0.298 | −0.254 | 0.000 |
| T3 | 0.133 | −0.316 | 0.000 | 0.000 | 0.258 | 0.597 | −0.182 | 0.000 |
| T4 | −0.114 | −0.434 | −0.400 | 0.000 | −0.116 | 0.143 | 0.402 | 0.180 |
| T5 | −0.321 | −0.326 | 0.000 | −0.403 | 0.000 | −0.151 | 0.218 | −0.395 |
| T6 | −0.383 | −0.202 | 0.190 | 0.121 | −0.263 | 0.138 | −0.506 | −0.154 |
| T7 | −0.428 | 0.000 | 0.159 | 0.102 | 0.000 | 0.000 | 0.167 | 0.638 |
| T8 | −0.287 | 0.345 | 0.223 | 0.172 | 0.105 | 0.000 | 0.314 | −0.437 |
| T9 | −0.131 | 0.411 | −0.377 | 0.161 | 0.000 | −0.130 | −0.310 | −0.179 |
| T10 | 0.000 | 0.396 | −0.138 | −0.372 | 0.118 | 0.253 | −0.107 | 0.289 |
| T11 | 0.241 | 0.221 | −0.123 | 0.000 | −0.340 | 0.448 | 0.213 | −0.208 |
| T12 | 0.192 | 0.000 | −0.321 | −0.202 | −0.481 | −0.285 | 0.000 | 0.000 |
| T13 | 0.239 | 0.000 | 0.388 | −0.490 | −0.168 | −0.192 | 0.000 | 0.000 |
| T14 | 0.263 | −0.158 | 0.250 | 0.339 | −0.408 | 0.000 | −0.220 | 0.000 |
| Explained Variance | 0.101 | 0.096 | 0.084 | 0.083 | 0.082 | 0.073 | 0.068 | 0.066 |

| Variables | Components | | | | | | |
|---|---|---|---|---|---|---|---|
| | PC9 | PC10 | PC11 | PC12 | PC13 | PC14 | PC15 |
| T0 | 0.206 | 0.447 | 0.439 | 0.209 | 0.183 | 0.263 | 0.114 |
| T1 | 0.000 | 0.370 | −0.495 | 0.169 | 0.000 | −0.357 | 0.000 |
| T2 | 0.466 | −0.259 | 0.000 | 0.144 | −0.231 | 0.395 | 0.000 |
| T3 | 0.277 | 0.122 | 0.000 | −0.362 | −0.150 | −0.341 | 0.242 |
| T4 | 0.147 | 0.122 | −0.128 | −0.181 | 0.385 | 0.349 | −0.225 |
| T5 | 0.169 | 0.000 | 0.000 | 0.377 | 0.209 | −0.240 | 0.354 |
| T6 | 0.140 | 0.408 | 0.000 | 0.240 | −0.125 | 0.166 | −0.320 |
| T7 | 0.315 | 0.000 | −0.194 | 0.000 | −0.281 | 0.000 | 0.323 |
| T8 | 0.414 | −0.187 | 0.000 | −0.316 | 0.000 | 0.000 | −0.328 |
| T9 | 0.149 | 0.268 | 0.000 | −0.179 | 0.353 | 0.171 | 0.469 |
| T10 | 0.235 | 0.000 | −0.165 | 0.306 | 0.344 | −0.293 | −0.353 |
| T11 | 0.000 | 0.000 | −0.299 | 0.314 | −0.316 | 0.332 | 0.235 |
| T12 | 0.350 | 0.198 | 0.416 | −0.105 | −0.293 | −0.233 | −0.119 |
| T13 | 0.000 | 0.197 | −0.454 | −0.451 | 0.000 | 0.156 | 0.000 |
| T14 | 0.333 | −0.449 | 0.000 | 0.000 | 0.404 | −0.127 | 0.101 |
| Explained Variance | 0.058 | 0.057 | 0.056 | 0.051 | 0.048 | 0.037 | 0.031 |

The eigenvalue distribution of the embedding matrix of the series generated by $N(0,1)$, by the very weak added signal, is not completely flat (as in the ideal case of pure noise),

but shows a slow decay from the 10.1% explained variance of PC1 to the 3.1% of PC15. This result is consistent with the character of the correlation filter of PCA, exalting any correlation structure present in the data. The plot reported in Figure 1 suggests a five-component solution as the bona fide signal according to the LEV criterion [20], setting the signal/noise border at the reaching of a linear relation between the logarithm of the variance explained and the number of components.

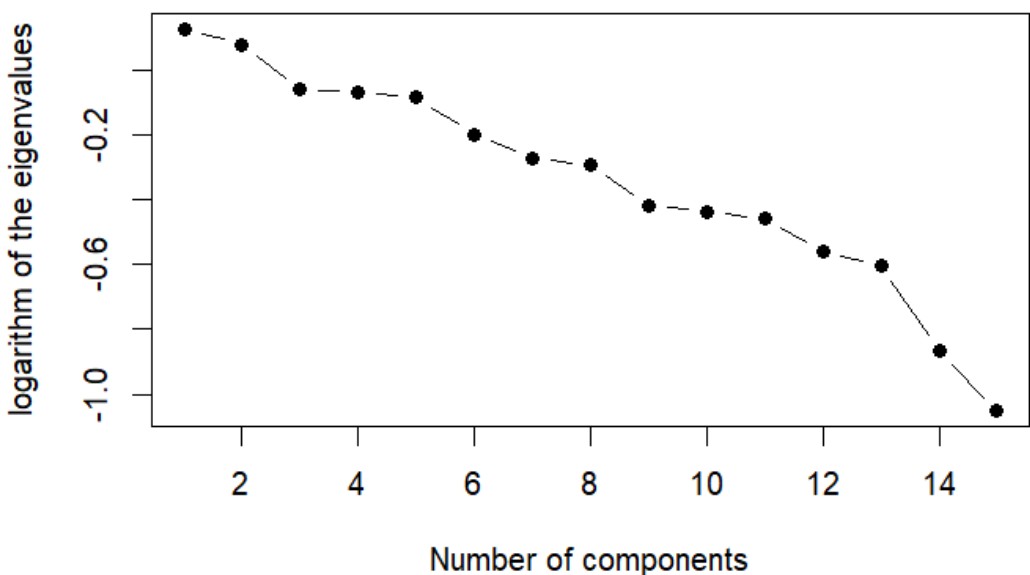

**Figure 1.** LEV diagram of the signal. The logarithm of the eigenvalues is shown on the h-axis. The number of principal components is chosen accordingly to the value of *y*, for which the graph assumes, approximately, a straight-line trend. In this case, the number of bona fide signal components suggested by the method was five.

The saw-wave signal (alternating $0.1/-0.1$ values) (see Section 2) does not correspond to the 'signal components' as selected by the LEV criterion, but is embedded in PC14, well inside the noise floor. PC14's loadings show a clear resemblance to the added saw-like alternating weak signal with alternating opposite sign loadings (the only exceptions being T7, T8, and T9, which in turn have the lowest absolute loading values on PC14).

The addition of a saw-wave of alternating $0.1/-0.1$ values (see Section 2) to $S0$ did not alter the original signal (original series = $S0$; original series + low signal = $S1$), the original and signal-added series showing a Pearson r = 0.994 (Figure 2). Consequently, the $A0$ and $A1$ matrices are practically identical.

The 20 noise-contaminated replicas of the $S1$ series (with the amount of Gaussian added noise going from $\sigma = 0.1$ to $\sigma = 1$) were transformed into 20 multivariate matrices ($C1$–$C20$) by a 15-dimensional unit lag embedding and analyzed by PCA (see Section 2). The Pearson correlation coefficient between the corresponding noise floor components of the original ($A1$) and noise-corrupted ($C1$–$C20$) matrices were computed. As previously underlined in the Methods Section, the fact that PC6–PC15 constitute the noise floor implies that their eigenvalues are almost coincident. This, in turn, provokes the alteration of their ordering in the noise-added $C1$–$C20$ matrices. Thus, for any value of added noise, we picked the component having the higher correlation with the original one as the 'corresponding components', independent of the relative order of explained variance. The last step is the computations of the $R^2$ between the amount of added noise and the PC6–PC15 original/noisy pairs. The entire procedure going from the generation of noise-corrupted series and the $R^2$ of the relation between the amount of added noise and the

correlation between the original/noisy component pairs was iterated 100 times starting from different seeds. The $R^2$ relative to each iteration were eventually averaged, obtaining the following values for PC6–PC15 (Table 3).

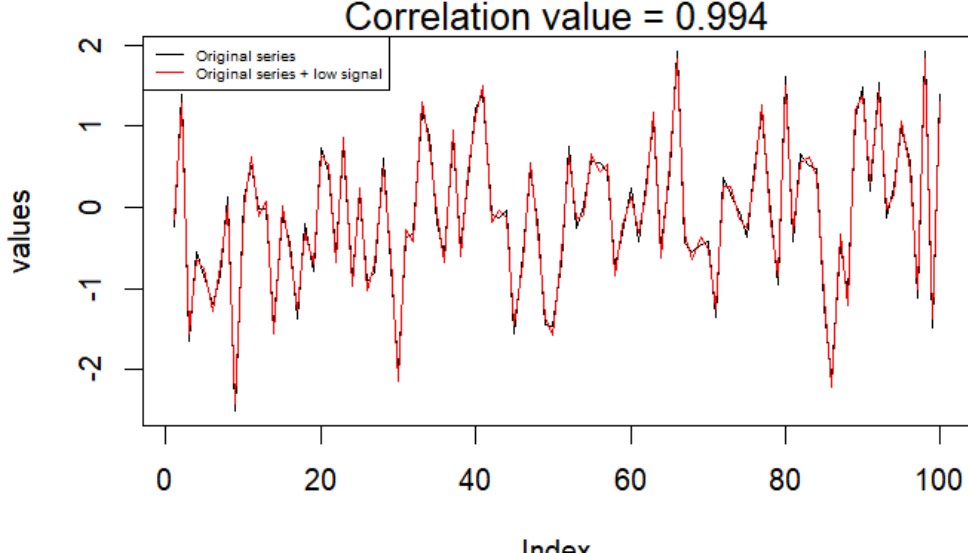

**Figure 2.** The X-axis shows the sequence of values subsequently extracted in order; the Y-axis shows the simulated values for the original series $S0$ (black line) and the original series + small signal $S1$ (red line). The two series are practically coincident and score a Pearson correlation near unity. The weak signal is almost perfectly masked by the Gaussian noise.

**Table 3.** The table reports the adjusted $R^2$ of the relation between the original/noise-corrupted correlation and the amount of added noise in the case of the 'noise floor' (PC6–PC15) components. It is worth noting the lack of any evident link between the component ordering and correlation values and the decidedly higher sensitivity to the added noise of PC14 with respect to the other components.

| Value of Adjusted $R$-Squared | | | | | | | | | |
|------|------|------|------|------|------|------|------|------|------|
| PC6 | PC7 | PC8 | PC9 | PC10 | PC11 | PC12 | PC13 | PC14 | PC15 |
| 0.393 | 0.588 | 0.350 | 0.530 | 0.612 | 0.662 | 0.609 | 0.600 | 0.800 | 0.681 |

As expected, PC14, mirroring the added weak signal, had a higher $R^2$ despite its extremely low eigenvalue. The average $R^2$ for PC1–PC16 was 0.58, with a standard deviation equal to 0.13, and the confidence interval at 99% was [0.45–0.72], thus pointing to a statistically significant departure of PC14 from the other minor components. This result, far from being a definitive proof of a general methodology to select putative signal components in a unsupervised way, seems to be a promising avenue to follow, to complement the more-reliable supervised approach based on the correlation with external variables [10].

### 4. Conclusions

The titration metaphor predicts the end of the 'reaction' when the analyte can no longer react with the added reactants: the stopping of the reaction comes from the total conversion of the analyte into the reaction product. In the case of noise titration, this implies that the relationship between the correlation of the corrupted signal with its original version arrives at a bottom end of near-zero correlation. Figure 3 depicts this condition.

The above result tells us that, analogous to chemical titration, even in the case of computational titration, it is of utmost importance to carefully select both the range of added noise and the dosage schedule. It is worth underlining that our procedure relies on the classical effects of noise in degrading self-correlated signals that (as demonstrated in [17]) do not need to be time series or to be generated by a specific function.

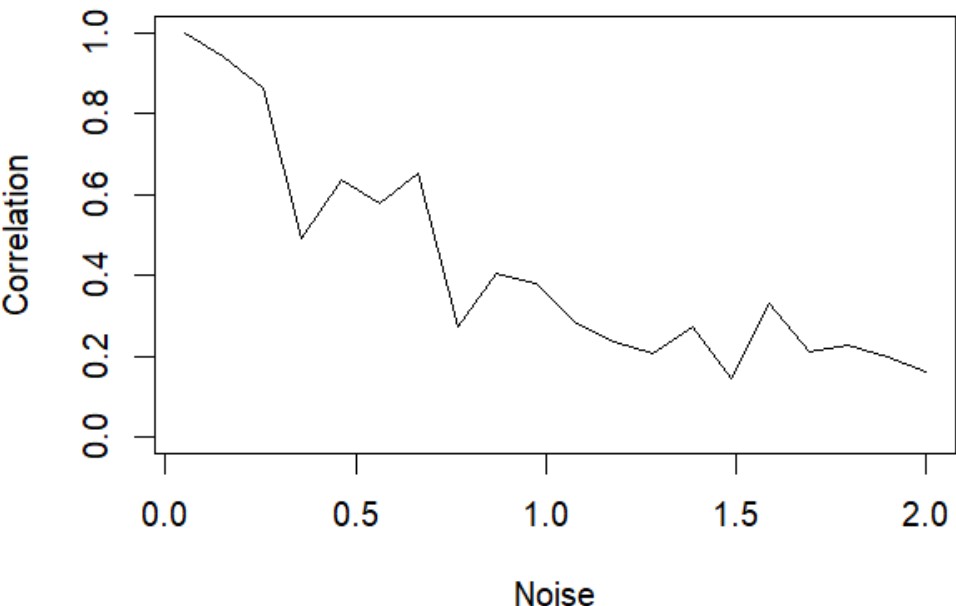

**Figure 3.** For increasing noise levels (X-axis), the correlation of the corrupted signal with its original version (Y-axis) reaches a bottom end of near-zero correlation. A unit value of added noise can be considered as the approximate 'volume of titration' for the analyte (signal) in the solution; when reaching this point, the linear correlation/added noise relation is lost.

To go beyond a purely empirical demonstration of the proposed methodology, we can refer to [21]. Formally, we wish to choose between the two hypothesis:

$$\mathcal{H}_0 : x[n] = w[n]$$
$$\mathcal{H}_1 : x[n] = A + w[n]$$

(1)

for $n = 0, \ldots, N - 1$ (Equation (1)). We have that $x$ is our data, $A$ is the known signal (saw signal), and $w$ is the Gaussian noise. In this case, the hypothesis $\mathcal{H}_0$ corresponds to the situation where the data are just noise, while the hypothesis $\mathcal{H}_1$ is the signal plus the Gaussian noise. A reasonable approach might be to average the samples and compare the value obtained to a threshold $\gamma$, so that we could decide $\mathcal{H}_1$ if

$$T(x) = \frac{1}{N} \sum_{n=0}^{N-1} x[n] > \gamma_x$$

(2)

where $T(x)$ is the statistic test. If $w[n]$ is a Gaussian mixture, the optimal likelihood ratio test (LRT) is used to decide if $\mathcal{H}_1$ is true. With a fixed probability of false alarm (level of significance $\alpha$) and, hence, threshold $\gamma$, it will produce the maximum probability of detection $P_D$ (power of the test). We can use another approach to find a better value for $P_D$. We assumed a realization of white Gaussian noise (WGN) with variance $\sigma^2$, which is independent of $w[n]$. Calling this $u[n]$, we have $y[n] = x[n] + u[n]$. It follows that we decide for $\mathcal{H}_1$ if

$$T(y) = \frac{1}{N} \sum_{n=0}^{N-1} y[n] > \gamma_y$$

(3)

In [21], it was proven that this approach provides a higher value for $P_D$ with respect to the statistical test $T(x)$. This result can be easily generalized to $R^2$.

The crucial issue of the paper of Kay [21] was the demonstration of the additive character of the noise ($w[n]$) with respect to the signal ($A$) evident in Equation (1). This corresponds to the titration metaphor in which the analyte ($A$) reacts with a reactant that corresponds to a different chemical ($w[n]$) added to the solution.

A last remark is related to the use of $R^2$ between the original and noise corrupted (hypothetical) signal instead of the 'titration volume' (i.e., the reaching of a flat correlation/added noise condition), like in analytical chemistry. At odds with chemistry, in which the goal of titration is the quantification of the concentration of an analyte in a solution, we are interested in a qualitative yes/no response about the signal character of a minor component. For this last task, the evidence of a reaction whose rate scales with the amount of added reactant rather than the precise amount of reactant saturating the initial analyte (volume of titration) is much more cogent. That is to say that the higher the $R^2$, the higher the tenability of $\mathcal{H}_1 = A + w[n]$ corresponding to a pre-existing analyte concentration ($A$) independent of the added reactant ($w[n]$).

It is worth noting that the same fundamental distinction between the signal and noise reported in Equation (1) holds for sparse PCA [22] in which the authors, as for the estimation of the 'real covariance matrix', depurated by the effect of noise, explicitly stated '... in practice, one does not have access to the population covariance, but instead must rely on a "noisy" version of the form $\Sigma = \Sigma + \Delta$, where $\Delta = \Delta_n$ denotes a random matrix perturbation'. Sparsity is frequent in biological data analysis, and in [22], the authors suggested an estimation strategy based on the search for the invariance of the PCA solution with different choices of reduced (and thus, perturbed) data sets. Sparsity is the focus of the interpretability score discussed in [11] and shares the same epistemological fundamentals of [22]: in order to interpret an emergent correlation structure (i.e., a component), we must be able to demonstrate that the component has a preferential link (e.g., correlation) with only one or at most a few variables, these being internal to the study [11] or external, as in [10]. In our work, we complemented this basic information theory concept [9] with a chemistry-inspired procedure relying on the hypothesis that a 'true' signal decays by its 'reaction' (titration) with added noise until it becomes no longer distinguishable by randomness (no preferential correlation).

As we stated before, the methodology we proposed in this paper can be of help in any situation in which we need to ascertain the signal character of a minor component in the absence of an external covariate. This implies that the proposed method could be adopted in both the feature selection and explanation procedures in machine learning [23]. Similar considerations hold in dynamical systems analysis, which, in the case of biological systems, often implies the simultaneous presence of different dynamical modes [24]. Besides the resemblance to already-established theoretical results and the wide range of applicability, what we considered the main motivation for this work was the need for statistical science to get back to its empirical roots. It is not by chance that the fathers of modern statistics (e.g., Ronald Fisher, Francis Galton) were biologists and their works were deeply inspired by the need to face peculiar biological phenomena. Here, we started from a very common situation encountered in experimental biology, i.e., the decoupling between the syntactical and semantical value of information mirrored by extremely strong signals (main component) devoid of any relevant semantic interest, going together with minor components in which semantically relevant information can hardly be distinguished from noise. We proposed an empirical solution to this problem based on a routine analytical chemistry method (titration), which holds promise to be of help in a wide range of situations.

**Author Contributions:** A.G. and A.V. contributed equally to the conceptualization, methodology, formal analysis, and writing—original draft preparation. All authors have read and agreed to the published version of the manuscript.

**Funding:** This research received no external funding.

**Institutional Review Board Statement:** Not applicable.

**Informed Consent Statement:** Not applicable.

**Data Availability Statement:** The original contributions presented in the study are included in the article. Further inquiries can be directed to the corresponding authors.

**Conflicts of Interest:** The authors declare no conflicts of interest.

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
