# Peer review of "On the (Apparently) Paradoxical Role of Noise in the Recognition of Signal Character of Minor Principal Components"

_stats, doi:10.3390/stats7010004_

Round 1

Reviewer 1 Report

Comments and Suggestions for Authors

The paper is not clear. What is your new contribution? Do you only review previous results?

Also, you should mention sparse PCA if you are talking about biloigcal data. Works like

"High-dimensional analysis of semidefinite relaxations for sparse principal components"

and 

"A greedy anytime algorithm for sparse PCA"

Comments on the Quality of English Language

NA

Reviewer 2 Report

Comments and Suggestions for Authors

According to my opinion, the manuscript is well-written and has merits for publication in the Journal. I therefore recommend publication of this work after some minor revisions based on the following:

1) Authors are encouraged to indicate again the main practical applications of this work.

2) The practical applications of this work should be more indicated.

3) For general readers, author is encouraged to discuss the possibility to use Machine learning models by discussing the following recent works: 

https://doi.org/10.12989/cac.2022.30.1.033

https://doi.org/10.1016/j.enganabound.2022.08.001

4) Figs.1 and 2 should be more discussed.

5) Conclusion section is poor. Some applications of the model and future scope should be included.

Comments on the Quality of English Language

The English is good.

Author Response

Please see the attachmnent

Reviewer 3 Report

Comments and Suggestions for Authors

After reading carefully I found the paper can be accepted. If the following changes can be done  

 1. In abstract the authors must highlight the motivation of the study.

2. What software was used for simulation?

3.  Improve all sections of the paper

4. The local search schemes should be used Levenberg propagation scheme instead of the active set scheme.

 5.  Improve the figures

6. Improve the conclusion

7.  Update all the references

  A stochastic computing procedure to solve the dynamics of prevention in HIV system, Biomedical Signal Processing and Control, 

  A numerical simulation of the fractional order Leptospirosis model using the supervise neural network, 

  Thin film flow and heat transfer of Cu-nanofluids with slip and convective boundary condition over a stretching sheet. 

 "Simulation of Thermal Decomposition of Calcium Oxide in Water with Different Activation Energy and the High Reynolds Number",

  designing a new fast solution to control isolation rooms in hospitals depending on artificial intelligence decision, The inclination of magnetic dipole effect and nanoscale exchange of heat of the Cross nanofluid, Partial differential equations modeling of bio-convective sutterby nanofluid flow through paraboloid surface. Effect of Ca doping on the arbitrary canting of magnetic exchange interactions in La1-xCaxMnO3 nanoparticles. Thermal and solute aspects among two viscosity models in synovial fluid inserting suspension of tri and hybrid nanomaterial using finite element procedure. Modeling and Computational Framework of Radiative Hybrid Nanofluid Configured By a Stretching Surface Subject To Entropy Generation: Using Keller Box Scheme 

8.  real application of the considered problem with the different mentioned effects should be mentioned and discussed.

9. The abstract should contain some quantitative information

10.  The results and discussion section should highlight more physical aspects of the present research.

Author Response

Please see the attacchment

Round 2

Reviewer 1 Report

Comments and Suggestions for Authors

You indeed improved the the presentation of the paper. I think that discussion about PCA and sparse PCA is still a bit lacking. 

Here are two works that discuss the interpretation of PCA, that may be interesting for you to follow on. I'd add them to the introduction discussion.

(1)"To interpret or not to interpret PCA? This is the question."

(2)“ Applications of principal component analysis integrated with GIS. ”

(3)“A principal component analysis (PCA)-based framework for automated variable selection in geodemographic classification.”

Comments on the Quality of English Language

please re-run a spell check or grammarly

Author Response

See file attached
